# SRPCA: Sparse Reverse of Principal Component Analysis for Fast Low-Rank Matrix Completion

## Abstract

Supervised and unsupervised learning methods experience a decline in performance when applied to incomplete, corrupted, or noisy datasets. Matrix completion is a common task to impute the missing values in sparsely observed matrices. Given a matrix $\mathbf{X} \in \mathbb{R}^{m \times n}$, low-rank matrix completion computes a rank-$r$ approximation of $\mathbf{X}$, where $r \ll \min\{m, n\}$, by only observing a few random entries of $\mathbf{X}$. It is commonly applied for recommender systems, image processing, and multi-output collaborative modeling. Existing matrix completion methods suffer either from slow convergence or failure under significant missing data levels. This paper proposes a novel approach, the Sparse Reverse of Principal Component Analysis (SRPCA), that reformulates matrix factorization based low-rank completion $(\min_{\mathbf{U}, \mathbf{V}} \|\mathcal{P}_{\mathbf{\Omega}}(\mathbf{X} - \mathbf{U}\mathbf{V}^T)\|_F^2)$ to iteratively learn a single low-rank subspace representation by solving the convex optimization problem $\min_{\mathbf{V}} \|\mathcal{P}_{\mathbf{\Omega}}(\mathbf{X} - \mathbf{P}\mathbf{V}^T)\|_F^2$ under the principal component analysis framework, resulting in a significant convergence acceleration. SRPCA converges iteratively and is computationally tractable with a proven controllable upper bound on the number of iterations until convergence. Unlike existing matrix completion algorithms, the proposed SRPCA applies iterative pre-processing resets that maintain smoothness across the reconstructed matrix, which results in a performance boost for smooth matrices. The performance of the proposed technique is validated on case studies for image processing, multivariate time-series imputation, and collaborative filtering. SRPCA is also compared with state-of-the-art benchmarks for matrix completion.

## 1 Introduction

Matrix completion is a common task for recovering missing or corrupted data in matrices (Wang & Fan, 2024). It has constantly received tremendous attention from many research fields such as collaborative filtering (e.g., recommender systems) (Yu et al., 2009; Chen & Wang, 2022), link analysis (Gleich & Lim, 2011), distance embedding (Candès & Recht, 2009), computer vision (Chen & Suter, 2004; Li et al., 2012), image processing (Ji et al., 2010; Jia et al., 2022), and so forth. In any field, missing data in high volumes has a negative impact on various data analysis processes, as many supervised and unsupervised learning methods cannot be applied directly to incomplete data (Audigier et al., 2016). Consequently, scalable and novel algorithms for matrix completion are still in constant demand, especially for applications with high levels of missing data.

The low-rank matrix has a key characteristic where the important information it contains, expressed in terms of degree of freedom, is significantly smaller than the total number of entries. This means that even if only a few entries are observed, there is still a good possibility of being able to reconstruct the entire matrix (Nguyen et al., 2019). Many data matrices analyzed are low-rank or approximately low-rank structured (Candès & Recht, 2009). Taking a movie recommender system as an example, there are only few factors that may contribute to users' preferences, suggesting that the data matrix recording users' rating scores is actually low-rank structured.

Most approaches that solve low-rank matrix completion problems can be mainly divided into two categories, nuclear norm based and matrix factorization based (Sun & Luo, 2016). In the first cate-

gory, the objective of matrix rank minimization is approximated by nuclear norm minimization. This category's methods include interior-point-based Semi-Definite Programming (SDP) solver (Candès & Recht, 2009), conjugate gradient method (Blanchard et al., 2015), Singular Value Threshold (SVT) algorithm (Cai et al., 2010), Augmented Lagrange Multiplier (ALM) algorithm (Lin et al., 2010), and robust principal component analysis (Zhang et al., 2012), etc. In the second category, the original matrix is compactly represented as the product of two low-rank matrices. The two low-rank matrices are usually iteratively updated through various algorithms such as Alternating Least Squares (ALS) (Jain et al., 2013; Gu et al., 2024) and Stochastic Gradient Descent (SGD) (Gemulla et al., 2011; Qin et al., 2024). Nuclear norm based algorithms are known to be more time-consuming as matrix dimension increases, while matrix factorization algorithms, which are non-convex heuristics, scale badly with high levels of missing data Gu et al. (2024). There is a very important requirement for a realistically and practically good matrix completion, which is the local and global smoothness in the reconstructed matrix. For example, in image processing, it is critical to obtain smoothness over the image. Often, this requirement comes at the cost of moderately updating the observed values, e.g., Gaussian filters and Variational Bayesian techniques (Kawasumi & Takeda, 2018; Paliwal et al., 2022).

To address this limitation, we propose the Sparse Reverse of the Principle Component Analysis (SRPCA) to complete matrices in their original space. The proposed approach maintains a high level of smoothness by iteratively finding the principal components of the matrix based on the predicted values of both the missing and the observed parts of the matrix, while guaranteeing that the principal components are capable of reconstructing the observed part of the matrix with minimal differences. The main contributions of the paper can be summarized in the following:

- A novel low-rank matrix completion method called SRPCA is proposed. It leverages both the missing and the observed part of the matrix-to-recover to iteratively learn the principle components that adequately represent the underlying low-rank matrix. SRPCA maintains smoothness across the reconstructed matric by applying an iterative pre-processing step.

- SRPCA is a matrix factorization based algorithm that modifies the non-convex problem to a convex one where it iteratively learns a single low-rank subspace representation, instead of two, by leveraging the principle component analysis framework. This aids in accelerating convergence.

- SRPCA is proved to improve in performance iteratively until convergence with a controllable upper bound on the number of iterations.

- An extension of SRPCA, called FastSRPCA, is proposed to offer a faster convergence in matrix completion applications where convergence rate is valued more than further improving a sub-optimal matrix recovery.

- SRPCA is evaluated on three diverse low-rank matrix completion based case studies and is shown to improve on other state-of-the-art benchmarks in terms of computational time and matrix recovery.

## 2 PRELIMINARIES

### 2.1 MATRIX COMPLETION

If we assume that the data matrix to be recovered has a low-rank structure, the matrix completion problem should be defined as follows (Candès & Recht, 2009):

$$\min_{\boldsymbol{M}} \text{rank}(\boldsymbol{M}), \quad \text{s.t. } X_{i,j} = M_{i,j}, \forall (i,j) \in \boldsymbol{\Omega} \tag{1}$$

where $\boldsymbol{X} \in \mathbb{R}^{m \times n}$ is the sparse observed matrix, $\boldsymbol{M} \in \mathbb{R}^{m \times n}$ the reconstructed matrix of $\boldsymbol{X}$, and $\boldsymbol{\Omega}$ represents the observed entries of $\boldsymbol{X}$.

This problem is a simple explanation of the low-rank matrix completion problem. Unfortunately, the rank minimization is NP-hard and has led researchers to propose different relaxations to solve the problem. Specifically, a commonly used convex relaxation for the rank is the nuclear norm, $\|\boldsymbol{M}\|_*$, which approximates problem (1) as (Candès & Recht, 2009; Sun & Luo, 2016; Hardt & Wootters, 2014):

$$\min_{\boldsymbol{M}} \|\boldsymbol{M}\|_*, \quad \text{s.t. } X_{i,j} = M_{i,j}, \forall (i,j) \in \boldsymbol{\Omega} \tag{2}$$

Or:

$$\min_{\boldsymbol{M}} \left[ \tau \left\| \boldsymbol{M} \right\|_* + \frac{1}{2} \left\| \boldsymbol{X} - \boldsymbol{M} \right\|_F^2 \right] \tag{3}$$

where $\left\| \cdot \right\|_F^2$ is the Frobenius norm and $\tau$ is apositive penalty parameter. Equations (2) & (3) can be conveniently optimized through some interior-point-method-based SDP solvers (Candès & Recht, 2009), like SDPT3 and SeDuMi. Cai et al. (2010) further proposed an SVT algorithm to solve (2) & (3). In the SVT, the estimate $\boldsymbol{M}$ converges to a unique solution of (3) through an iterative algorithm.Unfortunately, both SDP solvers and the SVT algorithm are problematic when applied to a large-size data set (Sun & Luo, 2016). Especially for the SVT, SVD computation is required at each iteration, which is time-consuming.

Another common approach to relax the rank is via matrix factorization, in which the unknown data matrix is expressed as the product of two low-rank matrices, $\boldsymbol{U}$ and $\boldsymbol{V}$ Sun & Luo (2016). In this case, the low-rank condition is satisfied automatically, and problem (1) can be transformed to:

$$\min_{\boldsymbol{U},\boldsymbol{V}} \left\| \mathcal{P}_{\boldsymbol{\Omega}}(\boldsymbol{X} - \boldsymbol{U}\boldsymbol{V}^T) \right\|_F^2, \quad \text{s.t. } \boldsymbol{M} = \boldsymbol{U}\boldsymbol{V}^T \tag{4}$$

Problem 4 is clearly a non-convex optimization problem. Compared with the nuclear norm based approach, the matrix factorization based approach performs much better on computation time. ALS is one of the popular matrix factorization-based methods, which originates from the power factorization method (Haldar & Hernando, 2009). In the ALS algorithm, the observed entries are randomly partitioned into a number of subsets at first. Then, $\boldsymbol{U}$ and $\boldsymbol{V}$ are initialized through the SVD of the first subset of the observed matrix. Next, at each iteration when moving to the next subset, $\boldsymbol{U}$ and $\boldsymbol{V}$ are alternatively updated to minimize the difference between $\boldsymbol{U}\boldsymbol{V}^T$ and the observed entries of that subset. ALS decreases the computational time because it does not apply SVD at each iteration. However, it may lead to high inaccuracies at high levels of missing data, and it ignores the smoothness of the data set due to the random partitioning of the original matrix. A recent matrix factorization based method is the Gauss-Newton Matrix Recovery (GNMR) (Zilber & Nadler, 2022), which utilizes a Gauss-Newton method to solve for the two factor matrices every iteration. Another method we use for benchmarking is Low-Rank Gaussian Copula (LRGC) (Zhao & Udell, 2020) which is a semiparametric algorithm for data imputation that also offers uncertainty quantification. For more comprehensive surveys, we refer readers Nguyen et al. (2019).

## 2.2 PRINCIPLE COMPONENT ANALYSIS (PCA)

PCA is one of the most widely used statistical tools for data analysis and dimensionality reduction (Candès et al. (2011)). It has been applied in many different areas, such as quantitative finance (Han et al., 2023; Chin et al., 2023), neuroscience (Lawrence et al., 2023), and image processing (Mishra et al., 2024). PCA provides a roadmap for transforming the original data set to a new basis with a lower dimension, thus filtering out the noise and revealing the hidden simplified dynamics. Therefore, with PCA, it is possible to extract critically important information from original data, thus simplifying the data structure.

Suppose we have a data matrix $\boldsymbol{M}$. The goal of PCA is to find an orthonormal matrix where $\boldsymbol{P} = \boldsymbol{M}\boldsymbol{V}$, such that the covariance matrix of $\boldsymbol{P}$ is diagonalized and expressed as:

$$\boldsymbol{S_P} = \frac{1}{n-1} \boldsymbol{P}^T \boldsymbol{P} \tag{5}$$

Since $\boldsymbol{P} = \boldsymbol{M}\boldsymbol{V}$, then:

$$\boldsymbol{S_P} = \frac{1}{n-1}(\boldsymbol{M}\boldsymbol{V})^T(\boldsymbol{M}\boldsymbol{V}) = \frac{1}{n-1}\boldsymbol{V}^T(\boldsymbol{M}^T\boldsymbol{M})\boldsymbol{V} \tag{6}$$

Let $\boldsymbol{V}$ be the eigenvectors matrix of $\boldsymbol{M}^T\boldsymbol{M}$; hence, matrix $\boldsymbol{S_P}$ is diagonalized. This is because $\boldsymbol{M}^T\boldsymbol{M} = \boldsymbol{V}\boldsymbol{D}\boldsymbol{V}^T$ and:

$$\boldsymbol{S_P} = \frac{1}{n-1}\boldsymbol{V}^T(\boldsymbol{V}\boldsymbol{D}\boldsymbol{V}^T)\boldsymbol{V} = \frac{1}{n-1}(\boldsymbol{V}^T\boldsymbol{V})\boldsymbol{D}(\boldsymbol{V}^T\boldsymbol{V}) = \frac{1}{n-1}\boldsymbol{D} \tag{7}$$

PCA is statistically intuitive and helps reduce the data's dimensions; however, applying PCA iteratively for matrix completion is time-consuming. Therefore, in this paper, we initialize the matrices $\boldsymbol{P}$ and $\boldsymbol{V}$ via PCA and efficiently update them via the proposed algorithm in the next section.

# 3 The Sparse Reverse of PCA (SRPCA)

## 3.1 Problem Formulation

Based on the general matrix completion problem, the goal is to construct a matrix $\boldsymbol{M} \in \mathbb{R}^{m \times n}$ that estimates the missing part of matrix $\boldsymbol{X} \in \mathbb{R}^{m \times n}$. Let $\boldsymbol{\Omega} = \{(i,j) : X_{i,j}$ is observed$\}$, $\mathcal{P}_{\boldsymbol{\Omega}}(\boldsymbol{X}) \in \mathbb{R}^{m \times n}$ to be the matrix that preserves the entities in $\boldsymbol{\Omega}$ and replaces the remaining entities by 0, and $\boldsymbol{\Omega}^{\perp}$ to be the complement of $\boldsymbol{\Omega}$. Contrary to the matrix factorization approach in 4, where the purpose is to consectively minimize for each of the two factor matrices, we minimize the problem to a convex problem, where we only solve for one of them as per the following with $\boldsymbol{M} = \boldsymbol{P}\boldsymbol{V}^T$:

$$\min_{\boldsymbol{P}} \left\| \mathcal{P}_{\boldsymbol{\Omega}}(\boldsymbol{X} - \boldsymbol{P}\boldsymbol{V}^T) \right\|_F^2 \tag{8}$$

where $\boldsymbol{P}$ is the principal component matrix, and $\boldsymbol{V}$ is the eigenvector matrix of $\boldsymbol{M}^T \boldsymbol{M}$.

## 3.2 Principal Components Estimation

To obtain the principal components, we first decompose the matrix $\boldsymbol{M}^T \boldsymbol{M} = \boldsymbol{V} \boldsymbol{\Lambda} \boldsymbol{V}^T$, where $\boldsymbol{\Lambda}$ is a diagonal matrix with $\lambda_j$ as its $j^{\text{th}}$ diagonal element, $\lambda_j$ is the $j^{\text{th}}$ eigenvalue of $\boldsymbol{M}^T \boldsymbol{M}$ corresponding to its $j^{\text{th}}$ eigenvector $\boldsymbol{V}_{:,j}$. Then, the principle components are estimated as:

$$\boldsymbol{P} = \boldsymbol{M}\boldsymbol{R}^T, \quad \text{where } \boldsymbol{R}^T = [\boldsymbol{V}_{:,\boldsymbol{r}}] \in \mathbb{R}^{n \times r} \tag{9}$$

with $\boldsymbol{r} \in \mathbb{R}^r$ being a vector of indices corresponding to the top $r$ eigenvectors. The selection of the top $r$ is for computational efficiency, compression and smoothing purposes. For matrix completion, $\boldsymbol{P}$ and $\boldsymbol{R}$ are updated iteratively.

## 3.3 The SRPCA Algorithm

The first step in the proposed approach is data standardization, which is common in data analytics:

$$\boldsymbol{X}_{:,j} = \frac{\boldsymbol{X}_{:,j} - \mu_{\boldsymbol{X}_{:,\boldsymbol{\Omega}_j}}}{\sigma_{\boldsymbol{X}_{:,\boldsymbol{\Omega}_j}} + \epsilon} \tag{10}$$

where $\boldsymbol{X}_{:,j}$ is the $j^{\text{th}}$ column of the matrix $\boldsymbol{X}$, $\mu_{\boldsymbol{X}_{:,\boldsymbol{\Omega}_j}}$ and $\sigma_{\boldsymbol{X}_{:,\boldsymbol{\Omega}_j}}$ are the mean and standard deviation of the available elements in the $j^{\text{th}}$ column $\boldsymbol{X}$, and $\epsilon$ is a small constant to avoid numerical instabilities when $\sigma_{\boldsymbol{X}_{:,\boldsymbol{\Omega}_j}} \to 0$.

A fair and an intuitive first approximation $\boldsymbol{M}^{(0)}$ is:

$$M_{i,j} = \begin{cases} X_{i,j} & \text{if } (i,j) \in \boldsymbol{\Omega} \\ \mathcal{N}(0,1) & \text{if } (i,j) \in \boldsymbol{\Omega}^{\perp} \end{cases} \tag{11}$$

Unlike many existing approaches, each iteration in SRPCA starts with $\boldsymbol{M}_{\boldsymbol{\Omega}}^{(k)} = \boldsymbol{X}_{\boldsymbol{\Omega}}$ because the observed values of $\boldsymbol{X}$ are unbiased estimates of the values in $\boldsymbol{\Omega}$. This serves a pre-processing reset from which all iterations start. Then, we proceed from (11) to obtain the new updates for $\boldsymbol{P}$ and $\boldsymbol{M}$:

$$\boldsymbol{P}^{(k)} = \boldsymbol{M}^{(k)} \boldsymbol{R}^{(k)^T} \tag{12}$$

$$\boldsymbol{M}^{(k+1)} = \boldsymbol{M}^{(k)} \boldsymbol{R}^{(k)^T} \boldsymbol{R}^{(k+1)} = \boldsymbol{P}^{(k)} \boldsymbol{R}^{(k+1)} \tag{13}$$

where $\boldsymbol{R}^{(k+1)}$ is iteratively returned by the algorithm.

Those updates conclude some major advantages of the proposed SRPCA so far:

(i) *Unbiased Estimate.* It starts with an unbiased estimate of the observed part of the matrix at every iteration. This is critical for scenarios with high percentages of missing data, because the first few iterative updates of the matrix are highly dependent on unreliable random prior estimates of the missing part of the matrix, which may slow the convergence or lead to divergent estimates of the matrix $\boldsymbol{M}$. Therefore, by keeping an unbiased estimate of the observed part of the matrix, it boosts the accuracy of the SRPCA to a certain extent.

(ii) *Smoothness.* The new update $M^{(k+1)}$ is smoother than the prior update $M^{(k)}$. Therefore, the SRPCA also helps to smoothen the original observed part of the matrix.

(iii) *Nonlinear Update.* The principal components are updated iteratively as shown in (13). This adds a layer of nonlinearity to the SRPCA.

As shown in (13), the update $M^{(k+1)}$ depends on the updated eigenvectors $R^{(k+1)}$. The SRPCA updates $R^{(k+1)}$ to maintain a certain level of accuracy for the observed data by solving for:

$$R^{(k+1)} = \underset{R^{(k+1)}}{\arg\min} \left\| \mathcal{P}_{\Omega}(X - P^{(k)} R^{(k+1)}) \right\|_F^2 \tag{14}$$

The objective function (14) serves two purposes by aiming to drive $\mathcal{P}_{\Omega}(X - P^{(k)} R^{(k+1)}) \to 0$: (i) it ensures a smooth transition from $M_{\Omega}^{(k+1)} = (P^{(k)} R^{(k+1)})_{\Omega}$ at the end of the $k^{\text{th}}$ iteration to $M_{\Omega}^{(k+1)} = X_{\Omega}$ at the beginning of the next $k+1$ iteration; and (ii) it quantifies and minimizes the differences between $M^{(k+1)}$ and the true matrix $X$. Thus, the updated $M^{(k+1)}$ is expected to provide a more realistic estimate of the missing data because now it provides a better estimate of the observed data.

Furthermore, because each column of $X$ can be expressed independently as a combination of the principal components, minimizing (14) is equivalent to the following convex optimization problem:

$$\underset{R_{:,j}^{(k+1)}}{\arg\min} (X_{:,j} - P^{(k)} R_{:,j}^{(k+1)})^T W^{(j)} (X_{:,j} - P^{(k)} R_{:,j}^{(k+1)}), \quad \forall j \tag{15}$$

where $W^{(j)} \in \mathbb{R}^{m \times m}$ is the weight matrix for the $j^{\text{th}}$ column of $X$ and it is a diagonal matrix such that $W_{i,i}^{(j)} = 1$ if $(i,j) \in \Omega$ and 0 otherwise. The sparse weight matrix $W^{(j)}$ provides all the weights corresponding to the observed values. Therefore, the solution of (15) is solely based on the observed part of $X$, and it can be written as per the following weighted least squares solution:

$$R_j^{(k+1)} = (P^{(k)^T} W^{(j)} P^{(k)})^{-1} P^{(k)^T} W^{(j)} X_{:,j}, \quad \forall j \tag{16}$$

Applying (16) is scalable for big data in the presence of parallel computation capabilities, allowing the simultaneous computation of different vectors of $R^{(k+1)}$ independently. Furthermore, the weight matrices are sparse and they do not require full matrix operations.

Finally, the algorithm converges when the improvement between two successive iterations is smaller than a predefined tolerance threshold. In other words, the algorithm terminates when $\left\| \mathcal{P}_{\Omega}(X - P^{(k-1)} R^{(k)}) \right\|_F^2 - \left\| \mathcal{P}_{\Omega}(X - P^{(k)} R^{(k+1)}) \right\|_F^2 \le \epsilon_{\text{tol}}$, where $M^{(k+1)} = P^{(k)} R^{(k+1)}$ and $\epsilon_{\text{tol}}$ is the tolerance threshold. Clearly, increasing $\epsilon_{\text{tol}}$ speeds up the algorithm convergence, but it also leads to a higher mean squared deviation $\left\| \mathcal{P}_{\Omega}(X - M^{(k+1)}) \right\|_F^2$. Therefore the choice of $\epsilon_{\text{tol}}$ depends on the application and the trade-off between speed and accuracy.

The pseudo-algorithm is demonstrated in Algorithm 1.

### 3.4 CONVERGENCE STUDY OF SRPCA

**Lemma 3.1** *The SRPCA converges iteratively with* $\left\| \mathcal{P}_{\Omega}(X - M^{(k+1)}) \right\|_F^2 \le \left\| \mathcal{P}_{\Omega}(X - P^{(k-1)} R^{(k)}) \right\|_F^2$. *Check Appendix A.1 for proof.*

Unlike some approaches in the literature, Lemma 3.1 shows that the performance of the SRPCA improves iteratively until it converges. This is a key finding, because if the algorithm terminates for external reasons (e.g., computational time constraints), the algorithm output will be the best-calculated estimate until the unexpected termination.

**Lemma 3.2** *The SRPCA converges at an iteration* $K < \left\lceil \frac{\left\| \mathcal{P}_{\Omega}(X - P^{(0)} R^{(1)}) \right\|_F^2}{\epsilon_{tol}} \right\rceil + 1$. *Check Appendix A.1 for proof.*

Lemma 3.2 provides an upper bound on the number of iterations until convergence, which also sets an upper bound on the computational time until convergence.

---

**Algorithm 1** The SRPCA for Matrix Completion.

---

1: $M_{i,j} = \begin{cases} X_{i,j} & \text{if } (i,j) \in \mathbf{\Omega} \\ \mathcal{N}(0,1) & \text{if } (i,j) \in \mathbf{\Omega}^\perp \end{cases}$      ▷ Data standardization and preprocessing

2: **for** $q \in [1, ..., n]$ **do**

3:     $\mathbf{W}^{(q)} = \mathbf{0}$      ▷ Construct the sparse weight matrices once

4:     $W_{i,i}^{(q)} = 1, \ \forall (i,q) \in \mathbf{\Omega}$

5: **end for**

6: $\mathbf{V}^{(0)}\mathbf{\Lambda}^{(0)}\mathbf{V}^{(0)^T} = \mathbf{M}^{(0)^T}\mathbf{M}^{(0)}$      ▷ Initial eigenvector decomposition.

7: $\mathbf{R}^{(0)} = [\mathbf{V}_{:,r}]^T, \ \mathbf{P}^{(0)} = \mathbf{M}^{(0)}\mathbf{R}^{(0)^T}, \ \mathbf{M}^{(1)} = \mathbf{M}^{(0)}$      ▷ Construct initial matrices

8: **for** $k \in [1, ..., maxIter]$ **do**

9:     Matrix smoothing      ▷ Optional

10:     $\mathbf{M}_{\mathbf{\Omega}}^{(k)} = \mathbf{X}_{\mathbf{\Omega}}$      ▷ Update for the the observed values

11:     $\mathbf{P}^{(k)} = \mathbf{M}^{(k)}\mathbf{R}^{(k)^T}$      ▷ Construct principal components

12:     **for** $j \in [1, ..., n]$ **do**

13:        $\mathbf{R}_j^{(k+1)} = (\mathbf{P}^{(k)^T}\mathbf{W}^{(j)}\mathbf{P}^{(k)})^{-1}\mathbf{P}^{(k)^T}\mathbf{W}^{(j)}\mathbf{X}_{:,j}$      ▷ Eigenvectors update

14:     **end for**

15:     $\mathbf{M}^{(k+1)} = \mathbf{M}^{(k)}\mathbf{R}^{(k)^T}\mathbf{R}^{(k+1)} = \mathbf{P}^{(k)}\mathbf{R}^{(k+1)}$      ▷ Update Matrix Estimate

16:     **if** $\left\|\mathcal{P}_{\mathbf{\Omega}}(\mathbf{X} - \mathbf{P}^{(k-1)}\mathbf{R}^{(k)})\right\|_F^2 - \left\|\mathcal{P}_{\mathbf{\Omega}}(\mathbf{X} - \mathbf{P}^{(k)}\mathbf{R}^{(k+1)})\right\|_F^2 \leq \epsilon_{\text{tol}}$ **then**

17:        Stop and Break      ▷ Stop when algorithm converges.

18:     **end if**

19: **end for**

---

## 3.5 EXTENSION: THE FAST SRPCA ALGORITHM

For many applications, the convergence rate is critical, and it is often acceptable to converge to solutions that are close enough to optimality. Recall that each iteration of the SRPCA starts with $\mathbf{M}_{\mathbf{\Omega}}^{(k)} = \mathbf{X}_{\mathbf{\Omega}}$ as a reliable unbiased estimate for the observed entities; however, this tends to slow down the convergence when $(\mathbf{P}^{(k-1)}\mathbf{R}^{(k)})_{\mathbf{\Omega}}$ is close but not equal to $\mathbf{X}_{\mathbf{\Omega}}$. Therefore, we propose the fast SRPCA (see Algorithm 2 in Appendix B), which starts each iteration with teh following pre-processing reset that is different than the original one in Algorithm 1:

$$\mathbf{M}_{\mathbf{\Omega}}^{(k)} = (1 - \alpha^*)(\mathbf{P}^{(k-1)}\mathbf{R}^{(k)})_{\mathbf{\Omega}} + \alpha^* \mathbf{X}_{\mathbf{\Omega}} \tag{17}$$

where $\alpha^* \in [0, 1]$ is a balancing scaler that is initialized to 1.

There are two main advantages for the choice of (17). First, $\alpha^*$ serves as a step-size because $\mathbf{M}_{\mathbf{\Omega}}^{(k)} = (\mathbf{P}^{(k-1)}\mathbf{R}^{(k)})_{\mathbf{\Omega}} + \alpha^*\left(\mathbf{X}_{\mathbf{\Omega}} - (\mathbf{P}^{(k-1)}\mathbf{R}^{(k)})_{\mathbf{\Omega}}\right)$; therefore, it is expected that $\alpha^* \to 0$ when $(\mathbf{P}^{(k-1)}\mathbf{R}^{(k)})_{\mathbf{\Omega}}$ is close enough to $\mathbf{X}_{\mathbf{\Omega}}$. Second, $\alpha^*$ serves as a smoothing parameter for noisy datasets where $\mathbf{X}_{\mathbf{\Omega}}$ is a noisy estimate for the observed entries $\mathbf{\Omega}$. For such noisy datasets, it is important to set $\alpha^* \to 0$ after enough iterations to avoid converging to a noisy estimate $\mathbf{M}_{\mathbf{\Omega}}^{(k)}$ that is close to $\mathbf{X}_{\mathbf{\Omega}}$.

**Lemma 3.3** *If $\alpha^* = 0$ at the beginning of iteration $K$, the fast SRPCA converges at iteration $K$ with $\mathbf{M}^{(K+1)} = \mathbf{M}^{(K)} = \mathbf{P}^{(K)}\mathbf{R}^{(K+1)} = \mathbf{P}^{(K-1)}\mathbf{R}^{(K)}$. Check Appendix A.2 for proof.*

From Lemma 3.3, it is intuitive to define $\alpha^*$ as a decreasing function with respect to the iteration number $k$. This speeds up the SRPCA convergence when $\left\|\mathcal{P}_{\mathbf{\Omega}}(\mathbf{X} - \mathbf{M}^{(k+1)})\right\|_F^2$ converges slowly to $\left\|\mathcal{P}_{\mathbf{\Omega}}(\mathbf{X} - \mathbf{P}^{(K-1)}\mathbf{R}^{(K)})\right\|_F^2$. However, a random choice of $\alpha^*$ may result in an unreliable estimate even for the observed part of the matrix with a large error $\left\|\mathcal{P}_{\mathbf{\Omega}}(\mathbf{X} - \mathbf{M}^{(k)})\right\|$. Thus, the choice of $\alpha^*$ depends on $\left\|\mathcal{P}_{\mathbf{\Omega}}(\mathbf{X} - \mathbf{M}^{(k)})\right\|$. Here, we propose $\alpha^*$ to be the solution for:

$$\arg\min_{\alpha}\left(\left\|\mathcal{P}_{\mathbf{\Omega}}(\mathbf{X} - \mathbf{M}^{(k)})\right\|_F + \delta|\alpha|\right) \tag{18}$$

where $\delta$ is a tuning parameter and for this specific choice of the objective function, it is the convergence threshold for $\left\|\mathcal{P}_{\mathbf{\Omega}}(\mathbf{X} - \mathbf{M}^{(k)})\right\|_F$ as shown in Lemma 3.4.

Table 1: The computational time and the full matrix recovery error $\left\|X - M^{(K+1)}\right\|_F^2$ for 50 replications at various missing data levels (50%, 70%, 80%) of a natural image (as seen in Figure 1).

| Method | Time (secs) | | | Full Reconstruction Error $(\times 10^{-3})$ | | |
|---|---|---|---|---|---|---|
| | 50% | 70% | 80% | 50% | 70% | 80% |
| **SRPCA** | **0.77 ± 0.034** | **0.91 ± 0.057** | **0.90 ± 0.040** | **7.0 ± 0.27** | **21.7 ± 0.58** | **40.2 ± 1.29** |
| ALM | 1.99 ± 0.014 | 1.95 ± 0.014 | 1.74 ± 0.032 | 19.5 ± 0.49 | 57.0 ± 1.64 | 79.3 ± 2.19 |
| SVT | 1.55 ± 0.014 | 3.43 ± 0.363 | 16.02 ± 0.580 | 19.5 ± 0.49 | 52.4 ± 3.68 | 134.9 ± 28.16 |
| ALS | 1.59 ± 0.020 | 1.57 ± 0.069 | 1.92 ± 0.580 | 43.1 ± 2.00 | 108.1 ± 6.09 | 395.0 ± 25.5 |
| LRCG | 31.84 ± 0.251 | 14.20 ± 0.608 | 9.55 ± 0.357 | 27.2 ± 0.65 | 73.1 ± 1.82 | 150.9 ± 5.41 |
| GNMR | 22.85 ± 1.933 | 7.17 ± 0.048 | 4.02 ± 0.153 | 38.8 ± 1.40 | 123.8 ± 4.37 | 309.7 ± 10.90 |

**Lemma 3.4** *The closed-form solution for (18) can be written as (proof in Appendix A.3):*

$$\alpha^* = \begin{cases} 0 & \text{if } \left\|\mathcal{P}_{\Omega}(X - M^{(k)})\right\|_F \leq \delta \\ 1 & \text{otherwise} \end{cases} \tag{19}$$

Lemma 3.4 shows that the fast SRPCA sets $M_{\Omega}^{(k)} = X_{\Omega}$ only when $P^{(K)}R^{(K+1)}$ does not accurately reconstruct the observed part of the matrix (i.e., when $\left\|\mathcal{P}_{\Omega}(X - M^{(k)})\right\|_F > \delta$). This also supports the validity of Lemmas 3.1 and 3.2 for the fastSRPCA, because for $\alpha^* = 1$ it becomes equivalent to the SRPCA and for $\alpha^* = 0$ it terminates at the same iteration.

## 4 EXPERIMENTATION AND RESULTS

We validate the efficacy of the SRPCA approach on case studies related to (i) image inpainting, (ii) multivariate time-series imputation, and (iii) collaborative filtering. All experiments were executed using the Intel Core i9-9980XE CPU with 62 GB RAM and done with Python 3.12.2, Numpy 1.26.4, SciPy 1.13.1

**Benchmark Models.** SRPCA is compared to state-of-the-art matrix completion methods: (i) inexact Augmented Lagrange Multiplier (ALM) (Lin et al., 2010), (ii) Single Value Thresholding (SVT) (Cai et al., 2010), (iii) Alternating Least Squares (ALS) (Duan, 2020), (iv) Low-Rank Gaussian Copula (LRGC) (Zhao & Udell, 2020), and (v) Gauss-Newton for Matrix Recovery (GNMR) (Zilber & Nadler, 2022). We note that not all benchmark models are designed to have the matrix's rank be a-priori appointed; the ones that do are ALS, LRGC and GNMR, along with SRPCA.

### 4.1 CASE STUDY 1: IMAGE INPAINTING

Images are often stored in the form of matrices, in which the intensity for pixel $(i, j)$ is stored in the matrix entry $(i, j)$. Furthermore, some pixels are often noisy or hard to obtain, and it is common to use matrix completion to reconstruct images. In this case study, a natural image of size $475 \times 344$ is used to validate the algorithm. Specifically, a uniform randomly selected subset – 50%, 70%, and 80% – of the pixels are removed and the matrix completion methods are then applied to reconstruct the image with $\epsilon_{tol} = 10^{-4}$.

**Results.** Figures 1, 2 and 3 show the outcomes of the matrix completion methods applied on a natural image with varying levels of missing data. While all the methods appear to decently recover the image at 50% missing level, the gap of the quality of reconstruction of that of SRPCA and the remaining methods increases as the level of missing data increases to 70% and 80%. This if further validated in Table 1, which shows the computational time required to reconstruct the image and the mean squared difference, $\|X - M\|_F^2$, of the standardized values of all entries of the matrix. The reconstruction error, for each method, increases with increase in missing data; this is natural because the matrix rank is probably underestimated with less observed data, leading to higher erros. SRPCA outperforms other methods in time and reconstruction error across all missing data levels, especially at large levels of 80% where other methods largely deteriorate. SRPCA considers the smoothness of the matrix and efficiently updates the principal components and eigenvectors in each iteration without explicitly running the eigenvector decomposition.

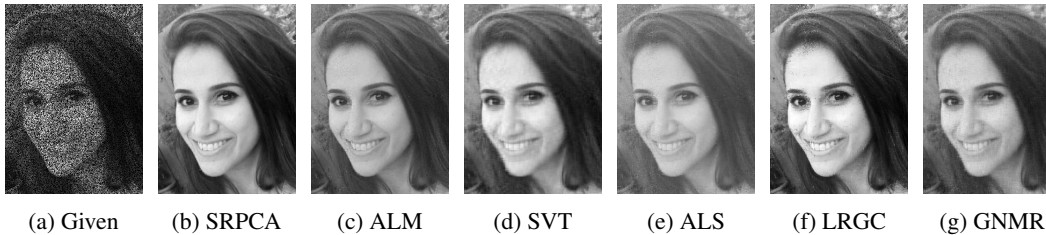

(a) Given    (b) SRPCA    (c) ALM    (d) SVT    (e) ALS    (f) LRGC    (g) GNMR

Figure 1: Reconstructed images with 50% missing pixels (the rank is set to 80).

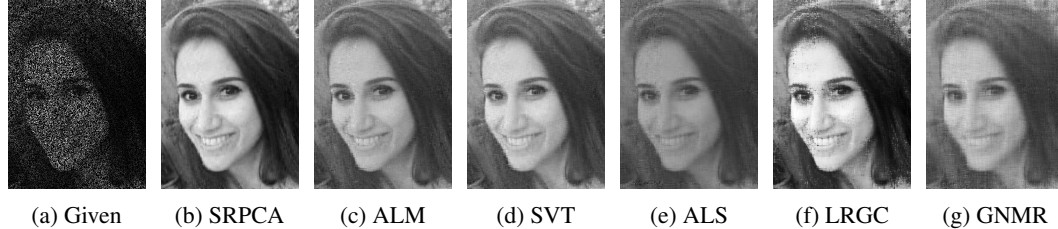

(a) Given    (b) SRPCA    (c) ALM    (d) SVT    (e) ALS    (f) LRGC    (g) GNMR

Figure 2: Reconstructed images with 70% missing pixels (the rank is set to 40).

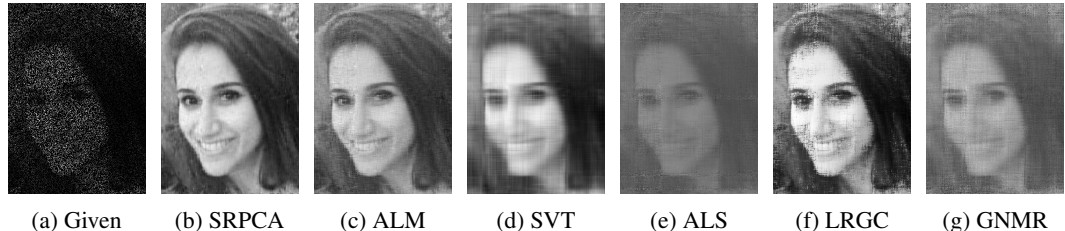

(a) Given    (b) SRPCA    (c) ALM    (d) SVT    (e) ALS    (f) LRGC    (g) GNMR

Figure 3: Reconstructed images with 80% missing pixels (the rank is set to 30).

Table 2: The computational time and the full matrix recovery error $\left\|X - M^{(K+1)}\right\|_F^2$ for 50 replications at different levels of missing data (10%, 30%, 50%, 70%) of turbine enignes dataset.

| Method | Time (secs) | | | | Full Reconstruction Error $(\times 10^{-3})$ | | | |
|---|---|---|---|---|---|---|---|---|
| | 10% | 30% | 50% | 70% | 10% | 30% | 50% | 70% |
| SRPCA | $2.11 \pm 0.047$ | $2.23 \pm 0.054$ | $2.44 \pm 0.042$ | $2.88 \pm 0.038$ | $\mathbf{31 \pm 0.3}$ | $\mathbf{96 \pm 0.5}$ | $\mathbf{170 \pm 0.8}$ | $286 \pm 2.4$ |
| ALM | $\mathbf{0.12 \pm 0.003}$ | $\mathbf{0.12 \pm 0.001}$ | $\mathbf{0.14 \pm 0.007}$ | $\mathbf{0.16 \pm 0.005}$ | $\mathbf{31 \pm 0.3}$ | $98 \pm 0.6$ | $\underline{181 \pm 4.7}$ | $\underline{279 \pm 3.0}$ |
| SVT | $0.70 \pm 0.022$ | $0.71 \pm 0.020$ | $0.71 \pm 0.015$ | $0.70 \pm 0.025$ | $264 \pm 0.3$ | $275 \pm 0.7$ | $296 \pm 1.2$ | $349 \pm 2.6$ |
| ALS | $4.16 \pm 0.265$ | $4.38 \pm 0.430$ | $-$ | $-$ | $263 \pm 0.3$ | $271 \pm 0.6$ | $-$ | $-$ |
| LRCG | $4.61 \pm 0.039$ | $4.59 \pm 0.041$ | $2.55 \pm 0.032$ | $8.44 \pm 0.058$ | $\underline{32 \pm 0.3}$ | $\underline{97 \pm 0.5}$ | $171 \pm 0.9$ | $\mathbf{276 \pm 2.4}$ |
| GNMR | $6.29 \pm 0.049$ | $5.52 \pm 0.139$ | $4.43 \pm 0.025$ | $3.22 \pm 0.013$ | $263 \pm 0.3$ | $271 \pm 0.6$ | $289 \pm 1.0$ | $345 \pm 2.4$ |

## 4.2 CASE STUDY 2: MULTIVARIATE TIME-SERIES IMPUTATION

Multivariate time series data frequently contains missing features with varying ratios and patterns depending on distinct sampling periods or measurement methods (Choi & Lee, 2024). These missing features can significantly impact downstream tasks. Time-series imputation is crucial in practical domains such as healthcare and prognostics. We evalaute the matrix completion algorithms on a multisensor monitoring data, in specific on a public aircraft gas turbine engines dataset (Saxena et al., 2008). The dataset contains measurements from 21 sensors recorded on 100 distinct engines that ran until failure. We focus on 12 sensors that show consistent trends across the 100 engines. Accordingly, the multisensor data from all 100 engines are stacked in one matrix with shape $20631 \times 12$. A uniform randomly selected subset – 10%, 30%, 50%, and 70% – of the matrix entries are removed and the matrix completion methods are then applied to recover the original matrix with $\epsilon_{tol} = 10^{-4}$. The rank for the underlying matrix is set to $r = 1$.

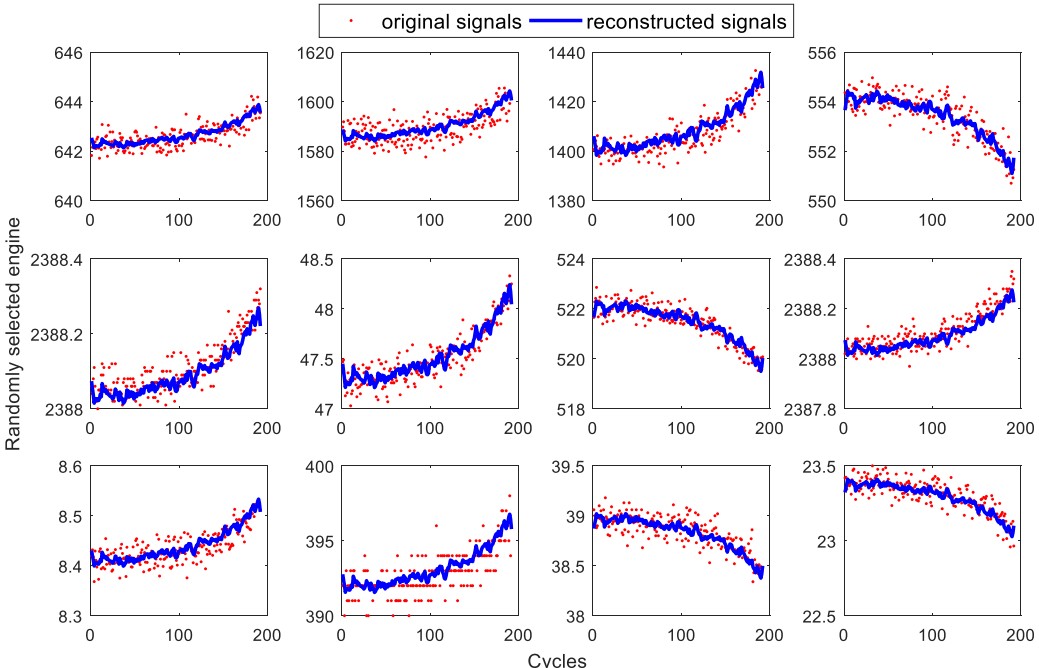

Figure 4: Reconstructed multisensor signals via the SRPCA approach with 70% missing data.

**Results.** The challenges in recovering this matrix lie in the fact that it (i) is already low-dimensional and (ii) has a high noise-to-signal ratio. This is evident in Figure 4 with the red scattered points of the original signal. The SRPCA reconstructs a filtered version of the signals eliminating a big part of the noise from the original. Table 2 shows the computational time required to reconstruct the signals and the mean squared difference, $\|X - M\|_F^2$, of the standardized values of all entries of the matrix. In terms of reconstruction, SRPCA, along with ALM and LRCG, produce the top matrix recoveries performances. Even though the low-dimensionality suits nuclear norm based methods like ALM, the SRPCA still produce top performances in acceptable timing. The ALS, on the other hand, diverges for high missing data levels of 50% and 70%.

### 4.3 CASE STUDY 3: COLLABORATIVE FILTERING

Collaborative filtering is a notable low-rank matrix completion application (Rennie & Srebro, 2005). It is one of the state-of-the-art techniques in recommender systems where the user-item interaction is embedded in matrix Li et al. (2021). Movie recommendations are a common recommender systems case study. The dataset we utilize for movie recommendations is the MovieLens 100k dataset that is available at https://grouplens.org/datasets/movielens/ (Harper & Konstan, 2015). The dataset contains 100k recommendations from 943 users for 1682 movies, which can be represented in a matrix of size 1682x943. The performance is evaluated by the normalized mean absolute error metric (NMAE) (Yang et al., 2018):

$$\text{NMAE} = \frac{\sum_{(i,j)\in\Omega^\perp} |M_{i,j} - X_{i,j}|}{(x_{\max} - x_{\min})|\Omega^\perp|} \tag{20}$$

where $x_{\max}$ and $x_{\min}$ are the maximum and minimum values, respectively, of the recommendation ratings. The rank of the underlying matrix is set to $r = 4$ and the threshold to $\epsilon_{tol} = 10^{-3}$. The matrix in this dataset poses different challenges than the previous too; it is heavily sparse in nature where only 100k entries are available, and that is because users tend to rate a select few movies. We analyze scenarios where 20%, 30%, 50% and 80% of the recommendations are randomly removed.

**Results.** Table 3 demonstrates the computational times and the reconstruction error of the missing entries (NMAE) for all methods across diverse missing data levels. It can be seen that SRPCA outperforms all other methods both in time to convergence and in the imputation of the missing

Table 3: The computational time and the normalized missing entries error NMAE for 50 replications at different levels of missing data (20%, 30%, 50%, 60%) of Movie100k dataset.

| Method | Time (secs) | | | | NMAE $(\times 10^{-3})$ | | | |
|---|---|---|---|---|---|---|---|---|
| | 20% | 30% | 50% | 60% | 20% | 30% | 50% | 60% |
| **SRPCA** | $9.64 \pm 0.086$ | $9.86 \pm 0.156$ | $9.89 \pm 0.856$ | $8.09 \pm 1.287$ | $180 \pm 1.1$ | $182 \pm 0.8$ | $189 \pm 7.0$ | $204 \pm 14.1$ |
| ALM | $17.78 \pm 1.281$ | $17.46 \pm 1.063$ | $17.02 \pm 2.718$ | $11.09 \pm 4.973$ | $221 \pm 3.3$ | $212 \pm 1.4$ | $207 \pm 6.6$ | $220 \pm 16.9$ |
| SVT | $14.82 \pm 6.165$ | $18.44 \pm 9.761$ | $30.16 \pm 7.861$ | $15.97 \pm 10.403$ | $278 \pm 90.6$ | $408 \pm 106.0$ | $513 \pm 7.2$ | $467 \pm 14.6$ |
| ALS | $14.44 \pm 2.339$ | $14.12 \pm 2.712$ | $15.69 \pm 3.040$ | $17.56 \pm 3.474$ | $190 \pm 1.6$ | $195 \pm 1.9$ | $208 \pm 2.3$ | $219 \pm 2.1$ |
| LRCG | $20.64 \pm 0.114$ | $21.28 \pm 0.885$ | $16.16 \pm 2.391$ | $13.97 \pm 9.171$ | $262 \pm 1.8$ | $273 \pm 1.6$ | $307 \pm 2.2$ | $322 \pm 5.8$ |
| GNMR | $11.97 \pm 0.254$ | $11.58 \pm 0.345$ | $10.11 \pm 0.832$ | $9.89 \pm 0.566$ | $218 \pm 11.3$ | $243 \pm 1.7$ | $355 \pm 15.6$ | $406 \pm 9.4$ |

data. This is a particularly huger dataset than the previous two case studies with much more intrinsic sparsity and extremely low rank. SRPCA leverages the smoothness in estimate update to converge faster than other methods. Taking advantage of iteratively resetting observed matrix entries, SRPCA achieves a lower error in recovering the unobserved ones.

## 5 CONCLUSION

Missing or corrupted data is pervasive across various fields, thus, affecting downstream learning tasks. Low-rank matrix completion is a common task to recover the missing entries of a partially observed matrix. Existing methods either fail under significant missing data levels or suffer from slow convergence. Th paper proposes a novel low-rank matrix completion method for incomplete datasets, called the Sparse Reverse of the Principle Component Analysis (SRPCA). The contributions are multi-fold: (i) this approach maintains a certain level of smoothness across the matrix; (ii) it modifies the non-convex matrix factorization problem into a convex optimization problem with a closed for solution, leveraging subspace representation in terms of principal components; (iii) it converges iteratively, which is critical for scenarios that stop the algorithm after reaching the maximum number of iterations or due to external reasons; and (iv) it is also computationally tractable, with a controlled upper bound on the number of iterations until convergence. In addition, a faster extension – fastSRPCA – is provided to improve on convergence rate at the expense of marginal missing data recovery. The efficacy of the SRPCA algorithm is validated on a natural image, multisensor dataset, and movie ratings dataset.

**Limitations.** (i) The first obvious limitation of SRPCA is that it requires the knowledge of the rank as an input. In most cases, this information is not readily available. However, there are low-rank matrix completion methods in literature (Nguyen et al., 2019) that we can influence from to make the rank selection adaptive. (ii) SRPCA suffers from singularity results in random predicitons for a column that has all its entries missing, which happens scarcely or for extremely high missing levels of data ( 90%). (iii) SRPCA is based on an assumption that the matrix completion environment is based on Missing-Completely-At-Random (MCAR). It is interesting for future studies to see how to develop it for Missing-At-Random (MAR) and Missing-Not-At-Random (MNAR) environments.

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

# A  PROOFS OF LEMMAS

## A.1  LEMMAS 3.1 AND 3.2

This appendix provides proofs for Lemmas 3.1 and 3.2. First, the rationale of adding $\boldsymbol{M}_{\boldsymbol{\Omega}}^{(k)} = \boldsymbol{X}_{\boldsymbol{\Omega}}$ is expected to iteratively improve the performance for scenarios where $\left\|\mathcal{P}_{\boldsymbol{\Omega}}(\boldsymbol{X} - \boldsymbol{P}^{(k-1)}\boldsymbol{R}^{(k)})\right\|_F^2$ is large. Intuitively, the algorithm should terminate when $\left\|\mathcal{P}_{\boldsymbol{\Omega}}(\boldsymbol{X} - \boldsymbol{P}^{(k-1)}\boldsymbol{R}^{(k)})\right\|_F^2 - \left\|\mathcal{P}_{\boldsymbol{\Omega}}(\boldsymbol{X} - \boldsymbol{P}^{(k)}\boldsymbol{R}^{(k+1)})\right\|_F^2 \leq \epsilon_{\mathrm{tol}}$, meaning when the error term $\left\|\mathcal{P}_{\boldsymbol{\Omega}}(\boldsymbol{X} - \boldsymbol{P}^{(k-1)}\boldsymbol{R}^{(k)})\right\|_F^2$ is not significantly decreasing anymore.

Next, we show that $\left\|\mathcal{P}_{\boldsymbol{\Omega}}(\boldsymbol{X} - \boldsymbol{P}^{(k-1)}\boldsymbol{R}^{(k)})\right\|_F^2 - \left\|\mathcal{P}_{\boldsymbol{\Omega}}(\boldsymbol{X} - \boldsymbol{P}^{(k)}\boldsymbol{R}^{(k+1)})\right\|_F^2 \leq \epsilon_{\mathrm{tol}}$ will keep decreasing until it becomes smaller than $\epsilon_{\mathrm{tol}}$. Specifically, at each iteration $k$, it is either:

$$\left\|\mathcal{P}_{\boldsymbol{\Omega}}(\boldsymbol{X} - \boldsymbol{P}^{(k-1)}\boldsymbol{R}^{(k)})\right\|_F^2 \leq \left\|\mathcal{P}_{\boldsymbol{\Omega}}(\boldsymbol{X} - \boldsymbol{P}^{(k)}\boldsymbol{R}^{(k+1)})\right\|_F^2 + \epsilon_{\mathrm{tol}} \tag{21}$$

Or:

$$\left\|\mathcal{P}_{\boldsymbol{\Omega}}(\boldsymbol{X} - \boldsymbol{P}^{(k)}\boldsymbol{R}^{(k+1)})\right\|_F^2 < \left\|\mathcal{P}_{\boldsymbol{\Omega}}(\boldsymbol{X} - \boldsymbol{P}^{(k-1)}\boldsymbol{R}^{(k)})\right\|_F^2 - \epsilon_{\mathrm{tol}} \tag{22}$$

For the first case in (21), the algorithm terminates at iteration $k$ by satisfying $\left\|\mathcal{P}_{\boldsymbol{\Omega}}(\boldsymbol{X} - \boldsymbol{P}^{(k-1)}\boldsymbol{R}^{(k)})\right\|_F^2 - \left\|\mathcal{P}_{\boldsymbol{\Omega}}(\boldsymbol{X} - \boldsymbol{P}^{(k)}\boldsymbol{R}^{(k+1)})\right\|_F^2 \leq \epsilon_{\mathrm{tol}}$.

For the second case, (22) leads to $\left\|\mathcal{P}_{\boldsymbol{\Omega}}(\boldsymbol{X} - \boldsymbol{P}^{(k)}\boldsymbol{R}^{(k+1)})\right\|_F^2 < \left\|\mathcal{P}_{\boldsymbol{\Omega}}(\boldsymbol{X} - \boldsymbol{P}^{(k-1)}\boldsymbol{R}^{(k)})\right\|_F^2$ because $\epsilon_{\mathrm{tol}} > 0$, and we move to the next iteration $k + 1$. Afterwards, similarly, either the SR-PCA terminates at $k + 1$ or $\left\|\mathcal{P}_{\boldsymbol{\Omega}}(\boldsymbol{X} - \boldsymbol{P}^{(k+1)}\boldsymbol{R}^{(k+2)})\right\|_F^2 < \left\|\mathcal{P}_{\boldsymbol{\Omega}}(\boldsymbol{X} - \boldsymbol{P}^{(k)}\boldsymbol{R}^{(k+1)})\right\|_F^2 - \epsilon_{\mathrm{tol}} < \left\|\mathcal{P}_{\boldsymbol{\Omega}}(\boldsymbol{X} - \boldsymbol{P}^{(k)}\boldsymbol{R}^{(k+1)})\right\|_F^2 < \left\|\mathcal{P}_{\boldsymbol{\Omega}}(\boldsymbol{X} - \boldsymbol{P}^{(k-1)}\boldsymbol{R}^{(k)})\right\|_F^2$.

This concludes the fact that $\left\|\mathcal{P}_{\boldsymbol{\Omega}}(\boldsymbol{X} - \boldsymbol{P}^{(k)}\boldsymbol{R}^{(k+1)})\right\|_F^2$ decreases iteratively until $\left\|\mathcal{P}_{\boldsymbol{\Omega}}(\boldsymbol{X} - \boldsymbol{P}^{(k-1)}\boldsymbol{R}^{(k)})\right\|_F^2 - \left\|\mathcal{P}_{\boldsymbol{\Omega}}(\boldsymbol{X} - \boldsymbol{P}^{(k)}\boldsymbol{R}^{(k+1)})\right\|_F^2 \leq \epsilon_{\mathrm{tol}}$ is satisfied. This concludes the proof for Lemma 3.1.

Next, we show that there exists an iteration $K$ such that $\left\|\mathcal{P}_{\boldsymbol{\Omega}}(\boldsymbol{X} - \boldsymbol{P}^{(k-1)}\boldsymbol{R}^{(k)})\right\|_F^2 - \left\|\mathcal{P}_{\boldsymbol{\Omega}}(\boldsymbol{X} - \boldsymbol{P}^{(k)}\boldsymbol{R}^{(k+1)})\right\|_F^2 \leq \epsilon_{\mathrm{tol}}$. Assume that the algorithm did not converge at iteration $K - 1$; therefore,

$$\begin{aligned}
0 &\leq \left\|\mathcal{P}_{\boldsymbol{\Omega}}(\boldsymbol{X} - \boldsymbol{P}^{(k-1)}\boldsymbol{R}^{(k)})\right\|_F^2 \\
&< \left\|\mathcal{P}_{\boldsymbol{\Omega}}(\boldsymbol{X} - \boldsymbol{P}^{(k-2)}\boldsymbol{R}^{(k-1)})\right\|_F^2 - \epsilon_{\mathrm{tol}} \\
&< \left\|\mathcal{P}_{\boldsymbol{\Omega}}(\boldsymbol{X} - \boldsymbol{P}^{(k-3)}\boldsymbol{R}^{(k-2)})\right\|_F^2 - 2\epsilon_{\mathrm{tol}} \\
&< \dots \\
&< \left\|\mathcal{P}_{\boldsymbol{\Omega}}(\boldsymbol{X} - \boldsymbol{P}^{(0)}\boldsymbol{R}^{(1)})\right\|_F^2 - (K-1)\epsilon_{\mathrm{tol}}
\end{aligned} \tag{23}$$

However, for those inequalities to hold, we must have $\left\|\mathcal{P}_{\boldsymbol{\Omega}}(\boldsymbol{X} - \boldsymbol{P}^{(0)}\boldsymbol{R}^{(1)})\right\|_F^2 - (K-1)\epsilon_{\mathrm{tol}} > 0$. Thus, the algorithm will terminate at an iteration $K$ such that:

$$K < \left\lceil \frac{\left\|\mathcal{P}_{\boldsymbol{\Omega}}(\boldsymbol{X} - \boldsymbol{P}^{(0)}\boldsymbol{R}^{(1)})\right\|_F^2}{\epsilon_{\mathrm{tol}}} \right\rceil + 1 \tag{24}$$

which satisfies $\left\|\mathcal{P}_{\boldsymbol{\Omega}}(\boldsymbol{X} - \boldsymbol{P}^{(K-1)}\boldsymbol{R}^{(K)})\right\|_F^2 - \left\|\mathcal{P}_{\boldsymbol{\Omega}}(\boldsymbol{X} - \boldsymbol{P}^{(K)}\boldsymbol{R}^{(K+1)})\right\|_F^2 \leq \epsilon_{\mathrm{tol}}$. This concludes the proof for Lemma 3.2.

## A.2   Lemma 3.3

This appendix proves Lemma 3.3. If $\alpha^* = 0$ at the beginning of iteration $K$, then the solution for:

$$\underset{\boldsymbol{R}_{:,j}^{(K+1)}}{\arg\min} (\boldsymbol{X}_{:,j} - \boldsymbol{P}^{(K)} \boldsymbol{R}_{:,j}^{(K+1)})^T \boldsymbol{W}^{(j)} (\boldsymbol{X}_{:,j} - \boldsymbol{P}^{(K)} \boldsymbol{R}_{:,j}^{(K+1)}), \quad \forall j = 1, .., n \tag{25}$$

is:

$$\boldsymbol{R}_{:,j}^{(K+1)} = (\boldsymbol{P}^{(K)^T} \boldsymbol{W}^{(j)} \boldsymbol{P}^{(K)})^{-1} \boldsymbol{P}^{(K)^T} \boldsymbol{W}^{(j)} \boldsymbol{X}_{:,j}, \quad \forall j = 1, .., n \tag{26}$$

This is also the solution for:

$$\underset{\boldsymbol{R}_{:,j}^{(K+1)}}{\arg\min} (\boldsymbol{X}_{:,j} - \boldsymbol{M}^{(K)} \boldsymbol{R}^{(K)^T} \boldsymbol{R}_{:,j}^{(K+1)})^T \boldsymbol{W}^{(j)} (\boldsymbol{X}_{:,j} - \boldsymbol{M}^{(K)} \boldsymbol{R}^{(K)^T} \boldsymbol{R}_{:,j}^{(K+1)}), \quad \forall j = 1, .., n$$
$$\tag{27}$$

which, because $\alpha^* = 0$, can be written as:

$$\underset{\boldsymbol{R}_{:,j}^{(K+1)}}{\arg\min} (\boldsymbol{X}_{:,j} - \boldsymbol{P}^{(K-1)} \boldsymbol{R}^{(K)} \boldsymbol{R}^{(K)^T} \boldsymbol{R}_{:,j}^{(K+1)})^T \boldsymbol{W}^{(j)} (\boldsymbol{X}_{:,j} - \boldsymbol{P}^{(K-1)} \boldsymbol{R}^{(K)} \boldsymbol{R}^{(K)^T} \boldsymbol{R}_{:,j}^{(K+1)}),$$
$$\forall j = 1, .., n \tag{28}$$

Therefore,

$$\boldsymbol{R}^{(K)} \boldsymbol{R}^{(K)^T} \boldsymbol{R}_{:,j}^{(K+1)} = (\boldsymbol{P}^{(K-1)^T} \boldsymbol{W}^{(j)} \boldsymbol{P}^{(K-1)})^{-1} \boldsymbol{P}^{(K-1)^T} \boldsymbol{W}^{(j)} \boldsymbol{X}_{:,j}, \quad \forall j = 1, .., n$$
$$= \boldsymbol{R}_{:,j}^{(K)}, \quad \forall j = 1, .., n \tag{29}$$

Finally, multiplying both sides of the equation by $\boldsymbol{P}^{(K-1)}$:

$$\boldsymbol{P}^{(K-1)} \boldsymbol{R}^{(K)} \boldsymbol{R}^{(K)^T} \boldsymbol{R}_{:,j}^{(K+1)} = \boldsymbol{P}^{(K-1)} \boldsymbol{R}_{:,j}^{(K)}, \quad \forall j = 1, .., n \tag{30}$$

which, because $\alpha^* = 0$, can be consecutively developed into:

$$\boldsymbol{M}^{(K)} \boldsymbol{R}^{(K)^T} \boldsymbol{R}_{:,j}^{(K+1)} = \boldsymbol{M}_{:,j}^{(K)}, \quad \forall j = 1, .., n \tag{31}$$

$$\boldsymbol{P}^{(K)} \boldsymbol{R}_{:,j}^{(K+1)} = \boldsymbol{M}_{:,j}^{(K)}, \quad \forall j = 1, .., n \tag{32}$$

$$\boldsymbol{M}_{:,j}^{(K+1)} = \boldsymbol{M}_{:,j}^{(K)}, \quad \forall j = 1, .., n \tag{33}$$

Thus, the fast SRPCA, in Algorithm 2, converges at the end of iteration $K$ by satisfying the $\left\| \mathcal{P}_{\boldsymbol{\Omega}}(\boldsymbol{X} - \boldsymbol{P}^{(K-1)} \boldsymbol{R}^{(K)}) \right\|_F^2 - \left\| \mathcal{P}_{\boldsymbol{\Omega}}(\boldsymbol{X} - \boldsymbol{M}^{(K+1)}) \right\|_F^2 = 0 \leq \epsilon_{\text{tol}}$. This concludes the proof of Lemma 3.3.

## A.3   Lemma 3.4

This appendix proves Lemma 3.4. First we rewrite 18 as:

$$\min_{\alpha} \left( \left\| \mathcal{P}_{\boldsymbol{\Omega}} \left( \boldsymbol{X} - (1-\alpha) \boldsymbol{P}^{(k-1)} \boldsymbol{R}^{(k)} - \alpha \boldsymbol{X} \right) \right\|_F + \delta|\alpha| \right) \tag{34}$$

$$= \min_{\alpha} \left( \left\| (1-\alpha) \mathcal{P}_{\boldsymbol{\Omega}}(\boldsymbol{X} - \boldsymbol{P}^{(k-1)} \boldsymbol{R}^{(k)}) \right\|_F + \delta|\alpha| \right) \tag{35}$$

$$= \min_{\alpha} \left( |1-\alpha| \left\| \mathcal{P}_{\boldsymbol{\Omega}}(\boldsymbol{X} - \boldsymbol{P}^{(k-1)} \boldsymbol{R}^{(k)}) \right\|_F + \delta|\alpha| \right) \tag{36}$$

Knowing that $\left\| \mathcal{P}_{\boldsymbol{\Omega}}(\boldsymbol{X} - \boldsymbol{P}^{(k-1)} \boldsymbol{R}^{(k)}) \right\|_F > 0$ and $\delta > 0$, then, the solution will satisfy $0 \leq \alpha^* \leq 1$.

Specifically, if $\left\| \mathcal{P}_{\boldsymbol{\Omega}}(\boldsymbol{X} - \boldsymbol{P}^{(k-1)} \boldsymbol{R}^{(k)}) \right\|_F < \delta$, then:

$$\left\| \mathcal{P}_{\boldsymbol{\Omega}}(\boldsymbol{X} - \boldsymbol{P}^{(k-1)} \boldsymbol{R}^{(k)}) \right\|_F = \min_{\alpha} \left( |1-\alpha| \left\| \mathcal{P}_{\boldsymbol{\Omega}}(\boldsymbol{X} - \boldsymbol{P}^{(k-1)} \boldsymbol{R}^{(k)}) \right\|_F + \delta|\alpha| \right) \tag{37}$$

with $\alpha^* = 0$.

Otherwise, if $\left\|\mathcal{P}_{\mathbf{\Omega}}(\boldsymbol{X} - \boldsymbol{P}^{(k-1)}\boldsymbol{R}^{(k)})\right\|_F > \delta$, then:

$$\delta = \min_{\alpha} \left(|1 - \alpha| \left\|\mathcal{P}_{\mathbf{\Omega}}(\boldsymbol{X} - \boldsymbol{P}^{(k-1)}\boldsymbol{R}^{(k)})\right\|_F + \delta|\alpha|\right) \tag{38}$$

with $\alpha^* = 1$.

For the case $\left\|\mathcal{P}_{\mathbf{\Omega}}(\boldsymbol{X} - \boldsymbol{P}^{(k-1)}\boldsymbol{R}^{(k)})\right\|_F = \delta$, any value $0 \leq \alpha^* \leq 1$ is a valid solution, but we chose $\alpha^* = 0$ to speed up the convergence of the algorithm. This concludes the proof for Lemma 3.4.

## B FASTSRPCA ALGORITHM

We provide the pseudo-algorithm of FastSRPCA as below:

---

**Algorithm 2** The Fast SRPCA for Matrix Completion.

---

1: $M_{i,j} = \begin{cases} X_{i,j} & \text{if } (i,j) \in \mathbf{\Omega} \\ \mathcal{N}(0,1) & \text{if } (i,j) \in \mathbf{\Omega}^{\perp} \end{cases}$ ▷ Data standardization and preprocessing

2: **for** $q \in [1, ..., n]$ **do**

3: $\quad \mathbf{W}^{(q)} = \mathbf{0}$ ▷ Construct the sparse weight matrices once

4: $\quad W_{i,i}^{(q)} = 1, \ \forall (i,q) \in \mathbf{\Omega}$

5: **end for**

6: $\mathbf{V}^{(0)}\mathbf{\Lambda}^{(0)}\mathbf{V}^{(0)T} = \mathbf{M}^{(0)T}\mathbf{M}^{(0)}$ ▷ Initial eigenvector decomposition.

7: $\mathbf{R}^{(0)} = [\mathbf{V}_{:,\boldsymbol{r}}]^{T}, \mathbf{P}^{(0)} = \mathbf{M}^{(0)}\mathbf{R}^{(0)T}, \mathbf{M}^{(1)} = \mathbf{M}^{(0)}$ ▷ Construct initial matrices

8: **for** $k \in [1, ..., maxIter]$ **do**

9: $\quad$ Matrix smoothing if $\alpha^{*} \neq 0$ ▷ Optional

10: $\quad \mathbf{M}_{\mathbf{\Omega}}^{(k)} = (1 - \alpha^{*})(\mathbf{P}^{(k-1)}\mathbf{R}^{(k)})_{\mathbf{\Omega}} + \alpha^{*}\mathbf{X}_{\mathbf{\Omega}}$ ▷ Update for the observed values

11: $\quad \mathbf{P}^{(k)} = \mathbf{M}^{(k)}\mathbf{R}^{(k)T}$ ▷ Construct principal components

12: $\quad$ **for** $j \in [1, ..., n]$ **do**

13: $\quad\quad \mathbf{R}_{j}^{(k+1)} = (\mathbf{P}^{(k)T}\mathbf{W}^{(j)}\mathbf{P}^{(k)})^{-1}\mathbf{P}^{(k)T}\mathbf{W}^{(j)}\mathbf{X}_{:,j}$ ▷ Eigenvectors update

14: $\quad$ **end for**

15: $\quad \mathbf{M}^{(k+1)} = \mathbf{P}^{(k)}\mathbf{R}^{(k+1)}$ ▷ Update Matrix Estimate

16: $\quad \alpha^{*} = \begin{cases} 0 & \text{if } \left\|\mathcal{P}_{\mathbf{\Omega}}(\mathbf{X} - \mathbf{M}^{(k)})\right\| \leq \delta \\ 1 & \text{otherwise} \end{cases}$ ▷ Update step-size.

17: $\quad$ **if** $\left\|\mathcal{P}_{\mathbf{\Omega}}(\mathbf{X} - \mathbf{P}^{(k-1)}\mathbf{R}^{(k)})\right\|_{F}^{2} - \left\|\mathcal{P}_{\mathbf{\Omega}}(\mathbf{X} - \mathbf{P}^{(k)}\mathbf{R}^{(k+1)})\right\|_{F}^{2} \leq \epsilon_{\text{tol}}$ **then**

18: $\quad\quad$ Stop and Break ▷ Stop when algorithm converges.

19: $\quad$ **end if**

20: **end for**

---

## C    EXTENDED RESULTS

### C.1    DESIGN OF EXPERIMENTS

In Table 4, we show the hyperparameters utilized to obtain the results corresponding to both SRPCA and fastSRPCA for this paper across the three case study datasets.

Table 4: List of hyperparameters pertaining to the SRPCA and fastSRPCA across all dataset. $\delta$ only corresponds to fastSRPCA.

| Datasets | % - list | rank | $maxIter$ | $\epsilon$ | $\epsilon_{\text{tol}}$ | Optional Smoothing | Standardization (10) | $\delta$ |
|---|---|---|---|---|---|---|---|---|
| **Image** | $[50, 70, 80]$ | $[80, 40, 30]$ | 50 | $10^{-5}$ | $10^{-4}$ | True | True | 0.08 |
| **Multisensor** | $[10, 30, 50, 70]$ | 1 | 50 | $10^{-5}$ | $10^{-4}$ | False | True | 0.4 |
| **MovieLens** | $[20, 30, 50, 60]$ | 4 | 50 | $10^{-5}$ | $10^{-3}$ | False | True | 0.7 |

### C.2    FASTSRPCA RESULTS

We extend the results shown in the paper with the performances produced by the fastSRPCA across the three datasets introduced. Figure 5 shows the reconstruced matrix by fastSRPCA across three distinct missing data levels. The recovered images look qualitatively as good as those produced by SRPCA.

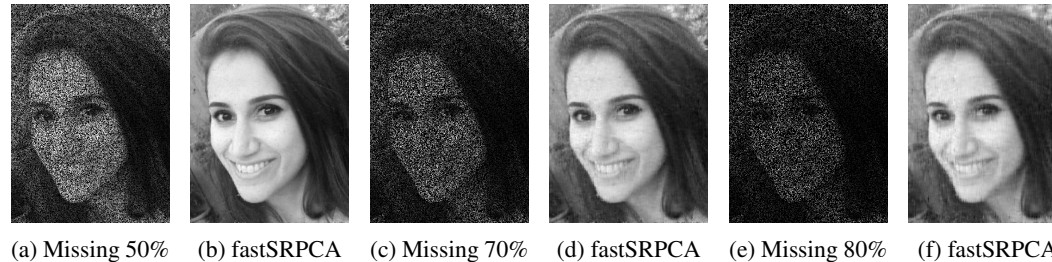

(a) Missing 50%    (b) fastSRPCA    (c) Missing 70%    (d) fastSRPCA    (e) Missing 80%    (f) fastSRPCA

Figure 5: Reconstructed images by fastSRPCA with diverse missing levels of pixels (the ranks are 80, 40 and 30 for 50%, 70% and 80%, respectively).

Table 5 shows the fastSRPCA computational time and matrix recovery performance in comparison to that of SRPCA for image dataset. While the performance is, as expected, secondary to that of SRPCA, the fastSRPCA still outperforms the other benchmarks shown in Table 1. Moreover, it converges faster than the SRPCA, almost taking hal the time SRPCA took for data missing levels of 50% and 70%.

Table 6 shows the fastSRPCA computational time and matrix recovery performance in comparison to that of SRPCA for the multisensor dataset. At low missing data levels, the fastSRPCA is capable of reaching the same or slightly worse matrix completion performance in less time. As the level of missing data increases, the gap between the two performances starts increasing.

Table 7 shows the fastSRPCA computational time and matrix recovery performance in comparison to that of SRPCA for the movie ratings dataset. The SRPCA was shown to be much faster than the other benchmarks in Table 3; however, the fastSRPCA is capable of converging in at least two seconds less across all missing data levels. Moreover, the matrix recovery descripency, demonstrated by the missing data error NMAE, between the two methods is small across all missing data levels.

Table 5: FastSRPCA's computational time and full matrix recovery error $\left\| X - M^{(K+1)} \right\|_F^2$ for 50 replications at various missing data levels (50%, 70%, 80%) of a natural image (as seen in Figure 1).

| Method | Time (secs) | | | Full Reconstruction Error $(\times 10^{-3})$ | | |
|---|---|---|---|---|---|---|
| | 50% | 70% | 80% | 50% | 70% | 80% |
| **SRPCA** | $0.77 \pm 0.034$ | $0.91 \pm 0.057$ | $0.90 \pm 0.040$ | $7.0 \pm 0.27$ | $21.7 \pm 0.58$ | $40.2 \pm 1.29$ |
| **fastSRPCA** | $0.35 \pm 0.009$ | $0.46 \pm 0.027$ | $0.68 \pm 0.034$ | $13.4 \pm 0.56$ | $27.4 \pm 1.05$ | $41.8 \pm 1.54$ |

Table 6: FastSRPCA's computational time and full matrix recovery error $\left\| X - M^{(K+1)} \right\|_F^2$ for 50 replications at different levels of missing data (10%, 30%, 50%, 70%) of turbine engines dataset.

| Method | Time (secs) | | | | Full Reconstruction Error $(\times 10^{-3})$ | | | |
|---|---|---|---|---|---|---|---|---|
| | 10% | 30% | 50% | 70% | 10% | 30% | 50% | 70% |
| **SRPCA** | $2.11 \pm 0.047$ | $2.23 \pm 0.054$ | $2.44 \pm 0.042$ | $2.88 \pm 0.038$ | $31 \pm 0.3$ | $96 \pm 0.5$ | $170 \pm 0.8$ | $286 \pm 2.4$ |
| **fastSRPCA** | $2.08 \pm 0.051$ | $2.12 \pm 0.053$ | $2.11 \pm 0.049$ | $2.19 \pm 0.037$ | $31 \pm 0.3$ | $97 \pm 0.7$ | $181 \pm 1.0$ | $314 \pm 5.4$ |

Table 7: The computational time and the normalized missing entries error NMAE for 50 replications at different levels of missing data (20%, 30%, 50%, 60%) of Movie100k dataset.

| Method | Time (secs) | | | | NMAE $(\times 10^{-3})$ | | | |
|---|---|---|---|---|---|---|---|---|
| | 20% | 30% | 50% | 60% | 20% | 30% | 50% | 60% |
| **SRPCA** | $9.64 \pm 0.086$ | $9.86 \pm 0.156$ | $9.89 \pm 0.856$ | $8.09 \pm 1.287$ | $180 \pm 1.1$ | $182 \pm 0.8$ | $189 \pm 7.0$ | $204 \pm 14.1$ |
| **fastSRPCA** | $6.98 \pm 0.049$ | $7.04 \pm 0.069$ | $7.09 \pm 0.145$ | $7.06 \pm 0.122$ | $187 \pm 1.2$ | $188 \pm 0.8$ | $194 \pm 5.9$ | $206 \pm 12.8$ |

