# OpenReview forum: "SRPCA: Sparse Reverse of Principal Component Analysis for Fast Low-Rank Matrix Completion"
_ICLR.cc/2025/Conference — ICLR 2025 Conference Withdrawn Submission_

### Official Review · Reviewer_Ahjx · 2024-10-29

**Soundness:** 2
**Presentation:** 1
**Contribution:** 1
**Rating:** 3
**Confidence:** 4

**Summary:**

This paper is focused on the matrix completion problem. It essentially employs a Burer-Monteiro factorization approach with alternating updates, after resetting the observed entries according to their given values at each iteration. A variant is proposed that interpolates between these observed values and the ones estimated by the algorithm, but the hyperparameter choice advocated by the authors effectively suppresses the interpolation.

**Strengths:**

A potential simplification of iterative schemes for matrix completion based on matrix factorization, where only one convex subproblem needs to be solved per iteration.

**Weaknesses:**

W1) While there seems to be some novelty in the way in which a factorization approach is employed by the algorithm (in which only one convex subproblem is solved per iteration instead of two), I do not think that the paper does a good job in motivating this technique (see W4 below) or brings useful insights into the studied problem. Furthermore, in my view the simple technique setting the observed entries to their given values along the iterations (see lines 201-202) is quite standard and not really an innovation.

W2) The paper claims multiple times to "maintain smoothness across the reconstructed matrix", but offers no concrete elements as to why this should be the case. In fact, based on what is promised multiple times (including in the abstract), one would expect a formalization of these ideas, and some sort of regularization or constraint being enforced for promoting smoothness. Instead, only quite vague and superficial comments are made, trying to imply that the simple, standard updates given in the paper are somehow capable of automatically promoting smoothness in the solution. In particular, in lines 216-217 it is said that the new estimate $M^{(k+1)}$ is "smoother" than $M^{(k)}$, but no formalization of what is meant, proof or even concrete heuristic arguments are given to support this claim.


W3) The paper suffers from a significant lack of clarity. Many sentences are poorly written, contain unusual expressions, improper formalization, or misinterpret basic notions such as convergence. Some examples are:
- "converges iteratively"
- "SRPCA is proved to improve in performance iteratively until convergence" (simply meaning that the objective function is non-increasing along iterations)
- Lemma 3.2 seems to mistake the convergence of an algorithm with its stopping. It claims to prove an upper bound on the number $K$ of iterations required for convergence, but it merely shows that the algorithm stops, as a consequence of the imposed stopping criterion.
- The solution to the problem in (14) is denoted with the same symbol as the optimization variable.

W4) The paper does not properly motivate its contents. It only vaguely mentions some standard difficulties faced by certain classes of algorithms, but does not explain in a clear way how exactly it proposes to overcome such difficulties. In particular, the alternating least-squares (ALS) approach is said to operate by randomly partitioning the observed entries into subsets (known as resampling), and then is criticized for ignoring "the smoothness of the data set due to the random partitioning of the original matrix." Yet, the authors are only focusing on a particular version of this approach: there is no reason why ALS must be employed with such a random partitioning in the first place (see, e.g., arXiv:1312.0925 and arXiv:1411.8003, the latter one containing a discussion on resampling).

W5) In the extension of Section 3.5, a parameter $\alpha^* \in [0,1]$ is introduced to interpolate the observed entries between their currently estimated and given values, instead of fixing them according to the latter. However, the authors propose a heuristic that amounts to setting $\alpha^* = 1$ until the difference (iin Frobenius norm) between the model and the observations is small on those entries, and then switching to $\alpha^* = 0$, at which point the algorithm is shown to stop. Therefore, the purported "smoothing" advantages brought by the interpolation are eliminated, and the introduced variant becomes equivalent to the previously proposed scheme.

W6) There are many typos along the text, indicating that it was not properly revised prior to submission.

**Questions:**

Q1) Why do you use the terminology "sparse reverse of PCA"?

Q2) Why should the new estimate $M^{(k+1)}$ be smoother than $M^{(k)}$, as claimed in lines 216-217, and in which sense exactly?

Q3) Line (13) is labeled "Eigenvectors update". It seems to refer to the eigenvectors of $(M^{(k)})^T M^{(k)}$, that is, the right singular vectors of $M^{(k)}$. But these should be orthogonal, and I don't see how the algorithm enforces this constraint. Otherwise, how is the update of $P^{(k)}$ (line 11 of Algorithm 1) supposed to work?

---

### Official Review · Reviewer_jUfs · 2024-11-03

**Soundness:** 3
**Presentation:** 3
**Contribution:** 3
**Rating:** 6
**Confidence:** 5

**Summary:**

In this paper, the author proposes a new matrix completion method that uses a PAC-like update regime. The convergence analysis is provided, and a fast version is developed. Experiment verifies the desired performance of the proposed method.

**Strengths:**

The method and theoretical analysis are technically sound. Experimental results show some advantages of the proposed method.

**Weaknesses:**

The time complexity is limited due to the utilization of SVD. The efficiency of large-scale data should be further investigated.

**Questions:**

1. What is the time complexity of the proposed method? Please add a time complexity analysis and provide the running time comparison in the experimental results section.
2. As the current experiments are small-scale data, how about the performance on large-scale data such as large images or MovieLens 1M? Will the time cost of the proposed method increase significantly?
3. The current experiment on image completion does not consider noisy situations. How about the performance of image completion with Gaussian noise?
4. Please compare the performance with the following work if possible: Li, Chao, Wei He, Longhao Yuan, Zhun Sun, and Qibin Zhao. "Guaranteed matrix completion under multiple linear transformations." In Proceedings of the IEEE/CVF Conference on Computer Vision and Pattern Recognition, pp. 11136-11145. 2019.

---

### Official Review · Reviewer_vB9k · 2024-11-03

**Soundness:** 3
**Presentation:** 2
**Contribution:** 2
**Rating:** 3
**Confidence:** 4

**Summary:**

This paper proposes a new iterative method for solving matrix completion problems in the presence of high levels of missing data. The proposed method is similar to ALS; the algorithm alternates between finding a least-squares solution (which has the benefit of being a convex problem) and performing a heuristic update. Theoretical claims of monotonic convergence of the iterates in finitely many steps are presented. A modified, faster-converging version of the algorithm is also presented. Finally, the authors provide numerical evidence of impressive performance on three datasets.

**Strengths:**

* The proposed methods (SRPCA and fastSRPCA) appear novel.
* The numerical results shown are promising. In particular, Figure 1-3 shows that SRPCA is significantly outperforming other methods, especially in the presence of large amounts of missing data.

**Weaknesses:**

* The key theoretical result (monotonic convergence to within epsilon tolerance in finitely many steps) is highly misleading. The termination/convergence criterion of the algorithm is previous_reconstruction_error - current_reconstruction_error < tol without absolute value bars. This means that as soon as an iterate has larger reconstruction error than the previous one, the algorithm will terminate. Trivially this means that the algorithm will monotonically decrease for as long as it runs precisely because it terminates as soon as monotonicity fails. However, this does not give any insight into whether the final iterate has, for example, good reconstruction error.
* While the numerical results shown are promising, they are limited overall. The key motivation is performance in the presence of limited data, so a phase transition plot indicating performance as function of amount of data present would be helpful. Further, the discussion focuses on how the algorithm converges very quickly, but no convergence plots are included. Finally, important parts of the experiments are not reproducible; the code provided in the supplement that creates Figures 1-3 does not run because of a missing file. (See suggestions below for details on what kinds of numerics would be helpful.)
* The presentation lacks clarity. The connection to ALS or alternating minimization methods is not made explicit in the text despite the fact that the algorithm can be interpreted as a variation of ALS. Specifically, the update to R(k) is solving a least-squares problem (i.e., one step of ALS). The update to P(k) is a heuristic update, but it is unclear what minimization problem this update is solving, if any.
* The “fast” variant of the proposed method (fastSRPCA) is poorly motivated. While the authors claim that it improves convergence at the cost of slightly degraded reconstructions, there is no empirical or theoretical comparison between fastSRPCA and SRPCA.
* Numerous other claims are unsubstantiated or false.
    * 197: Why is this a “fair and intuitive first approximation?”
    * 216: “The new update is smoother than the prior update.” This claim is never substantiated, nor is the relevant notion of smoothness defined.
    * 220: “This adds a layer of nonlinearity...” Where is the nonlinearity? All the iterative updates in SRPCA are matrix multiplications, which are linear.
    * 244: “Applying (16) is scalable for big data.” While the author points out the opportunity for parallelization, there is no analysis of the complexity of the algorithm with or without parallelization. This should be a straightforward calculation, and would lend credence to the fact that the algorithm could be scaled to large datasets.

**Questions:**

* Rephrasing the presentation in terms of the singular value decomposition would help elucidate what the updates correspond to. While an attempt is made at connecting R to an eigenvector matrix, after the first iteration R is no longer an eigenvector matrix and P no longer corresponds to principal components as traditionally defined. Further, the differences between the updates proposed and those performed in ALS should be clear.
* The authors claim that the proposed algorithm leads to “a performance boost for smooth matrices”. At a minimum, the authors should precisely define the notion of smoothness used.
* It would be helpful to have working code that fully reproduces Figures 1-3, including all comparisons to other methods.
* Could you make a phase transition plot indicating how performance degrades as a function of rank and proportion of missing data? See Figure 1 in https://arxiv.org/abs/0805.4471 and the associated description for the setup.
* One of the claims in the abstract is that the proposed method results “in a significant convergence acceleration“. To demonstrate this, the authors should include convergence plots depicting the reconstruction error of the algorithm (on the y-axis) as a function of the amount of iterations or time elapsed (on the x-axis), and repeat this for all methods compared in the experiments section. See Figure 1 in https://arxiv.org/abs/1605.07051 for an example.

Minor comments

* Line 110: apositive -> a positive
* Line 115: algorithm.Unfortunately -> algorithm. Unfortunately
* Line 119: you’re relaxing the minimization problem (1), not relaxing the rank
* Line 147-161: Variables are used before they are defined, including P, V, S_p, and n
* Line 168: entities -> entries
* Line 170: minimize the problem -> reduce the problem
* Line 11 in Algorithm 1: R(k) is used before it’s defined — probably should reorder lines 11-13.
* Line 13 in Algorithm 1: It is unclear that this matrix contains eigenvectors of any relevant matrix after the first update. It is also unclear that it will have orthonormal columns after the first update.
* Line 300: teh -> the
* Line 410: enignes -> engines
* Line 510: closed for -> closed form
* Line 519: “can influence from“ What does this mean?
* Line 520: predicitons -> predictions

---

### Official Review · Reviewer_5SVo · 2024-11-12

**Soundness:** 3
**Presentation:** 3
**Contribution:** 2
**Rating:** 5
**Confidence:** 3

**Summary:**

The paper introduces an iterative algorithm for matrix completion which the authors call Sparse Reverse Princial Component Analysis (SRPCA). The algorithm is based on iteratively (1) computing a PCA decomposition and (2) updating the principal vectors through a smooth gradient descent step.

**Strengths:**

The possibility to parallelize has some interest, the disadvantage being that you have to compute an eigenvalue decomposition at each step and to invert a matrix of size r by r (see (16)) which although not as computationally expensive as a full inverse can remain quite expensive.

**Weaknesses:**

My main concern has to do with the fact that the algorithm remains very close to existing algorithms (e.g. SVT from Cai et al. ) based on a combination of a gradient step and a truncated SVD. There exists an extensive body of work on matrix completion and any new approach (although perhaps interesting) would require a serious analysis and comparison with the literature. Had the authors shown a clear improvement in the rate of convergence (not just numerical as shown in Table 1) or in the number of samples needed for the recovery of the matrix, I would have been more inclined to accept the paper. But given that they only show convergence, I have to say I feel a little undecided. Perhaps the improvement should be better documented. E.g. what is the rate of convergence of SVT for example?

**Questions:**

A few additional comments:
 - One of the problems I have is that there already exist many approaches based on the SVD.
- Line 69/70 “Sparse Reverse of the Principle Component Analysis” —> “Principal Component Analysis”
- You should really simplify some of your introduction. E.g. “The proposed approach maintains a high level of smoothness by iteratively finding the principal components of the matrix based on the predicted values of both the missing and the observed parts of the matrix… ”. The sentence is way too long. I think I would remove the whole paragraph.
- I would remove at least 2 or 3 items from page 2 (lines 76 to 90) and give the algorithm. You mention your algorithm only vaguely by saying that it updates a single component. It would be a lot clearer to have a short, clear exposition of the algorithm (even informal). E.g. just give (12) - (14) with a few words of explanation.
- line 128/129, in your exposition of ALS, it is not clear why the entries are partitioned into subsets. That would be worth and explanation. Also when is the SVD computed. You say that it is not applied at each iteration. When is it applied then?
- line 136/137 “we refer readers ” —> “we refer readers to…”
- You should reorganize your explanation of PCA on lines 150 to 158. Start by something like “Let M denote any data matrix. Let V denote the eigenvectors of M^T M. We then define the matrix P as P = MV. The matrix P is diagonal. (You can then give (7)) ”.  You don’t need more than 2/3 lines to recall how PCA works
-  Lines 166 - 167, when you introduce P_\Omega, the proper technical term is “mask”.
- Lines 166-167 “the matrix that preserves the entities”, “and replaces the remaining entities” —> “entries”
- The notation “\Omega^\perp” is usually used to denote the orthogonal complement. I would use $\Omega^c$ for the complement.
- On lines 197-198, you first say that a fair intuitive approximation for M is (11) but then you seem to indicate that SRPCA starts with the restriction to the mask X_\Omega? Which initialization do you use? This is not clear. If you don’t use (11), this should go
- lines 226- 229 are not clear. Do you mean that the value of the matrix on the mask does not change? This does not seem true to me as when you compute the PCA decomposition you do not necessarily maintain the values on Omega. In any case, this should be rephrased.
- In your statement of Lemma 3.1., Why not replace P^{(k-1)}R^{(k)} by M^{(k+1)} ? Moreover, your statement does not imply convergence. The convergence would require a strict inequality (which I guess you have, yet just made a typo). Also the expression converges iteratively does not really mean anything. You can just use “converges” or in this case, you can even say “converges linearly with respect to the ||P_\Omega()||“ semi-norm
- line 262/263 “This is a key finding, because if the algorithm terminates for external reasons ” this is unneccesary
- line 270 - 292, in Algorithm 1, how do you determine the number of components that you retain. E.g. how do you set the value of r on line 278?
- The statement of Lemma 3.2. is unclear. What do you mean by “it converges at an iteration K”
- On Figure 2, the contrast does not seem to be the same between SRPCA and ALM
- line 300-301 “which start each iteration with teh” —> “with the”
- Lines 327 - 334, why are some of the convergence time and reconstruction errors underlined? You should clarify this.
- Line 312-313, the statement of Lemma 3.3. does not mean anything. What is K? From what I understand, K is defined with respect to a given tolerance epsilon_tol. This should be clarified.
- Line 312- 313, I find Lemma 3.3. somewhat misleading. From what I understand, the Fast RPCA algorithm corresponds to a change in the stopping criterion of Algorithm 1. I.e it basically tells you you should add the criterion ||P_{\Omega}(X - P^{(k-1)}R^{(k)})||_F^2< delta on top of the stopping criterion on the successive iterates. Why not merge this with algorithm 1? It would make sense to me to have a combined criterion of the form “if  deviation between iterates < epsilon  or error on Omega < delta  then stop“ It sort of makes sense that if your error on Omega get very small, you won’t be able to make further progress. You can keep part of the discussion from section 3.5 if you want but to me it would make more sense to merge the Fast SRPCA and the SRPCA algorithms (possibly with a short justification from section 3.5). This would be much clearer.

---

### Note · Authors · 2024-11-28

I have read and agree with the venue's withdrawal policy on behalf of myself and my co-authors.